# Association between Physical Activity and Dengue and Its Repercussions for Public Health: New Insights

**DOI:** 10.3390/ijerph21060727

**Published:** 2024-06-04

**Authors:** Francisco José Gondim Pitanga, Victor Keihan Rodrigues Matsudo

**Affiliations:** 1Postgraduate Program in Rehabilitation Sciences, Institute of Health Sciences, Federal University of Bahia (UFBA), Av. Reitor Miguel Calmon, s/n, Vale do Canela-, Salvador 40000-000, BA, Brazil; 2Physical Fitness Research Laboratory from São Caetano do Sul, CELAFISCS, São Caetano do Sul 01010-001, SP, Brazil; matsudo.celafiscs@gmail.com

**Keywords:** physical activity, the immune system, dengue, public health

## Abstract

Dengue is an endemic disease in tropical countries, mainly in South America, Southwest Asia, and Africa, which, despite having a low lethality rate, can overwhelm health systems. Strengthening the immune system through regular physical activity can be an important tool to prevent contagion, worsening, hospitalizations, and deaths caused by the disease, as seen in the COVID-19 pandemic. Therefore, this point of view aims to analyze the possible association between physical activity and dengue and its repercussions on public health. Comments were made on the main characteristics of dengue as well as on the main vaccines available to date. It was also discussed the impacts of dengue on health systems, in addition to the main repercussions for public health when a very large number of people are infected. It was also commented on the main factors that contribute to the worsening of the clinical stage of dengue, in addition to discussions and reflections on physical activity, strengthening the immune system, and dengue. There are assumptions that regular physical activity can be an important public health strategy to prevent contagion, severity, and hospitalizations caused by dengue and that it needs to be promoted by governments around the world as a tool for preventing and treating not only chronic communicable diseases but also infectious diseases.

## 1. Introduction

Dengue is an endemic disease in tropical countries, mainly in South America, Southwest Asia, and Africa [1]. From 2023 onwards, an unexpected increase in dengue cases around the world was observed, resulting in more than 6.5 million cases and 7300 deaths related to reported dengue cases [2]. In the Americas alone, during 2023, 4,565,911 cases of dengue were recorded [1]. This situation of high transmission continues in the year 2024, and specifically in Brazil; at the time of writing this document, 5,238,867 million cases already have been recorded, with 3038 deaths confirmed by this disease [3], with a consequent burden on the health system in some regions of the country.

Although vaccines already exist, they are not enough to immunize the entire population. It is noteworthy that the current Brazilian government is not managing to overcome the challenge, which is the fight against dengue.

On the other hand, strengthening the immune system through regular physical activity can be an important tool to prevent contagion, worsening, hospitalizations, and deaths caused by this disease, as seen in the COVID-19 pandemic [4,5,6,7,8,9,10].

Taking into account that dengue is an infectious disease, the mechanisms by which strengthening the immune system through regular physical activity, especially at moderate intensity, can help in prevention, as well as reducing the possibility of worsening the disease, seems to be quite plausible.

Considering that no manuscripts on physical activity and dengue were found in the consulted literature, this point of view proposes to comment on the new insights into the association between physical activity and dengue and its repercussions for public health.

## 2. What Is Dengue?

Dengue is an infectious disease transmitted by several species of mosquito of the *genus Aedes,* mainly *Aedes aegypti*. These mosquitoes also transmit zika and chikungunya. Dengue is widespread throughout the tropics, with local variations in risk depending on temperature, rainfall, and unplanned urbanization. There are four serotypes: DEN-1, DEN-2, DEN-3, and DEN-4. Infection with one of them provides permanent protection for the same serotype and partial and temporary immunity against the other three. Subsequent contagion by a different type of virus increases the risk of serious complications for the patient.

The dengue virus exists both in urban environments and in forest areas. In the urban environment, humans and mosquitoes are the only known hosts, while in forested areas, virus transmission occurs between non-human primates and, rarely, from these primates to humans [11].

After the mosquito bite and an intrinsic incubation period of 3 to 7 days, symptoms begin suddenly and follow three phases: an initial febrile phase, a critical phase around the defervescence period, and spontaneous recovery. Typically, patients go through the three phases without major complications, and in a small proportion of cases, typically in children and young adults, there is a systemic vascular leak that can worsen the disease [12].

## 3. Vaccines for Dengue

Although there are already vaccines developed in some countries around the world, in Brazil, they are still not enough to immunize the entire population. The existing vaccines to date are:

Dengvaxia, produced in France using attenuated virus technology. It is only recommended for people who have already been infected. In Brazil, it has been available since 2015, but only in private clinics.

Qdenga, developed in Japan, also uses attenuated virus technology. It can be applied to people who have already been infected or not. It was only made available in Brazil in February 2024, but not on a large scale, a fact that does not guarantee the immunization of the entire population at risk of the disease.

Butantan, developed in Brazil, also uses attenuated virus technology but is still in the final testing phase. Data with results from the follow-up of the first volunteers were recently published and demonstrated an efficacy of 79.6% [13].

## 4. The Impact of Dengue on Health Systems

Although dengue fever has a relatively low lethality rate (approximately 0.1% in the world [2] and 0.06% in the Brazil [3]), with the unexpected increase in cases around the world observed since 2023 and with a very large portion of the population infected, the demand for medical care becomes very high, a fact that can cause a very high demand for care, overloading health systems, as seen in the COVID-19 pandemic [14]. Furthermore, the cost of this high demand for care causes the evasion of resources that could be used on other fronts to combat diseases, mainly cardiometabolic diseases such as hypertension and diabetes, in addition to treatment for immunosuppressed, depression, and socially vulnerable people.

## 5. Factors That Can Prevent the Disease and/or Aggravate the Clinical Condition of Dengue

Different factors can contribute to the worsening of dengue; among them we can mention being younger, being female, having a high level of body mass index (BMI), the strain of the virus, in addition to variations in genetics related to the human major histocompatibility complex [12].

Among the factors mentioned above, the high level of BMI (obesity) is probably the one that deserves to be discussed in more depth, as there is a strong association between physical inactivity and obesity. Obesity is a risk factor for worsening dengue due to the inflammatory process triggered by it [12]. It is important to highlight that not only physical activity is important for preventing obesity, but also other behaviors such as healthy eating, hypertension, diabetes, a sedentary lifestyle, and difficulties in accessing primary healthcare in cases where mobility is also a constraint due to obesity.

Furthermore, the social and environmental conditions of the population must be observed. In this sense, managing environmental factors, such as avoiding the accumulation of water in containers indoors, cleaning urban areas, and using mosquito repellents, can be excellent strategies for preventing dengue fever.

On the other hand, other authors mention that mannose-binding lectin (MBL), a protein from the complement system of the innate immune system, can neutralize the dengue virus through mechanisms dependent and independent of the complement system itself, while its deficiency or polymorphisms may be an important component in the worsening of dengue [15,16]. Considering that it is suggested that physical activity may be associated with MBL, we will discuss this possible association in the next topic [17].

## 6. Physical Activity, Strengthening the Immune System, and Dengue

The immune system is made up of the innate defense system and the adaptive defense system. The innate defense system is made up of epithelial barriers, phagocytic cells, and natural killer cells, while the adaptive defense system is made up mainly of “B” lymphocytes, which produce antibodies, in addition to “T” lymphocytes that try to destroy infected cells.

Therefore, some researchers suggest that more physically active individuals could have more adequate defense mechanisms against infectious processes, mainly because they have higher concentrations of phagocytic cells as well as lymphocyte proliferation [4,18] (Figure 1).

Regarding the inflammatory process, seen mainly in obese individuals, the more physically active individuals could present more adequate defense mechanisms to reduce the inflammatory process caused by the conflict between the virus and our body’s immune cells through a greater concentration of anti-inflammatory substances, mainly interleukin 10 (IL-10), observed in greater quantities, chronically, in more physically active individuals [18].

Furthermore, changes or polymorphisms in MBL contribute to the activation of the complement system by recognizing dengue antigens. Complement system proteins are significantly modulated during the clinical course of dengue infection, suggesting that activation of the complement system through alternative pathways may be more dominant in the response to dengue infection and therefore may contribute to disease severity [15].

On the other hand, MBL can neutralize the dengue virus through mechanisms dependent and independent of the complement system [15,16], while physical activity can alter its levels. In a recent systematic review study, MBL values remained unchanged after an acute cycling session when compared to pre-exercise values. In this same study, the authors demonstrated an increase in MBL levels after long-duration running sessions [17].

MBL can restrict dengue infection via the recognition of high-mannose glycans in disease envelope proteins (15). Considering that physical activity can increase MBL levels, it is important to know the ideal intensity of physical exercise that can provide benefits for preventing and reducing the possibility of dengue worsening (Figure 1).

It is important to highlight that, although it has been shown that long-term exercise increases MBL levels, this same intensity and duration of physical activity can depress the immune system due to the “open window” effect observed after a series of long-term exercises [19]. Therefore, moderate-intensity exercises could probably be more suitable for strengthening the immune system.

## 7. Final Considerations

Considering the exponential increase in dengue cases observed since 2023 around the world, as well as the importance of combating the infectious process in addition to reducing its worsening, it becomes important to seek strategies to control the disease. Among important new insights, there is evidence that regular physical activity can be an excellent tool public health to prevent contagion, severity, and hospitalizations caused by infectious diseases, including dengue, and that it needs to be promoted by governments around the world as a strategy for preventing and treating not only chronic communicable diseases but also infectious diseases. Observational and intervention studies on the topic are suggested so that there can be further clarification on the possible association between physical activity and dengue and its consequent repercussions for public health.

## Figures and Tables

**Figure 1 ijerph-21-00727-f001:**
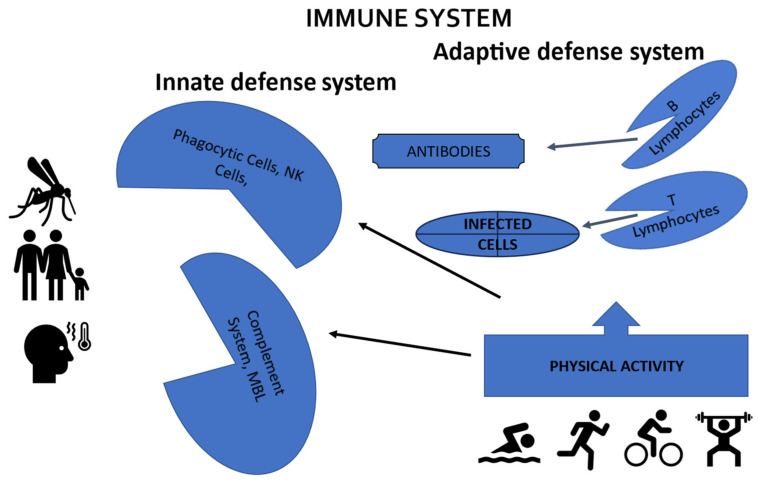
Strengthening the immune system through physical activity to prevent infection and the worsening of dengue symptoms.

## Data Availability

Not applicable.

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
