# Peer review of "Association between Physical Activity and Dengue and Its Repercussions for Public Health: New Insights"

_ijerph, 2024, doi:10.3390/ijerph21060727_

Round 1

Reviewer 1 Report

Comments and Suggestions for Authors

POSSIBLE ASSOCIATION BETWEEN PHYSICAL ACTIVITY AND DENGUE AND ITS REPERCUSSIONS FOR PUBLIC HEALTH

General comments:

The manuscript is a point of view that analyses the possible association between physical activity and dengue, and postulates regular physical activity as an important tool to prevent contagion, worsening, hospitalizations and deaths caused by the disease. The paper is very well written and contains relevant information to encourage reflection on the topic it proposes.

The authors focus the association of obesity with severe forms of dengue in terms of the physiological response of the immune system, and postulate physical activity as a solution. In the line of reasoning, they do not propose elements that are worldwide recognized to be associated with obesity, and that are fundamental to take into account such as eating habits, social vulnerability (poverty, problems of mobility to health care centers, etc.) and comorbidities generally associated with obesity such as diabetes, hypertension and others.

It is suggested to the authors to add a brief discussion that physical activity may not only be directly associated with the prevention and reduction of the risk of severe symptoms by strengthening the immune system, but may also be a result of an individual´s general health care behavior, taking care of their weight and body functionality, eating properly, preventing the spread of diseases, going for consultations with health professionals upon the appearance of abnormal symptoms in their body, etc.

Particular comments:

Lines 6-8. Revise Brazil (L 6) and Brasil (L 8).

L 17. Correct punctuation at the end of the sentence (..)

L 57. Aedes and Aedes aegypti  must be written in italics, and Chikungunya must be written in lower case.

L 65-67. It is suggested to rewrite the phrase ‘while in forested areas virus transmission occurs between nonhuman primates and, rarely, from these primates to humans’ to avoid confusion since, as written, it seems that dengue transmission is direct between primates and humans. On the other hand, this cycle in forested areas is probably sustained by another mosquito species, and only the Ae. aegypti mosquito has previously been mentioned as being responsible for urban dengue transmission, and it is suggested that this be clarified.

L 68. Clarify that this is the ‘instrinsic’ incubation period.

L 90. It is suggested to mention a percentage of the lethality rate globally and for Brazil.

L 94. It is suggested that you comment or list which are those other fronts to combat diseases to clarify this concept, and point out if it is a general observation or a particular one in Brazil.

L 98-100. It is suggested to list other relevant factors to contribute to the understanding of a broad public, beyond those that are intended to be emphasized in the context of this manuscript, such as certain comorbidities or pre-existing diseases (diabetes, hypertension, immunosuppressed), conditions of social vulnerability (access to the health system, access to care), pregnancy, pre-infection with a different serotype of dengue.

L 101-104. It is suggested to mention the relationship between obesity and other risk factors such as hypertension, diabetes, sedentary lifestyle and difficulties in accessing primary health care in cases where mobility is also a constraint due to obesity.

Figure 1. It is suggested that the cross symbol be changed so that the cell legend ‘infected cell’ can be read (if that is what it says, it is not really visible). Overall, this figure would not seem to represent or emphasize the concept to be highlighted. The authors could make an effort to improve it to make it visually self-descriptive.

Author Response

Dear Reviewer

Attached our answers.

Kind regards

Francisco Pitanga

Reviewer 2 Report

Comments and Suggestions for Authors

Dear authors,

First, congratulations on your excellent work. I would like to make some suggestions with the aim, from my humble knowledge, of improving the manuscript:

Title: In the title, I think that "possible" should be omitted since what is being studied is whether or not a relationship exists; starting with "possible" gives an image that even the authors are not very convinced of their work...

Abstract:

-          Line 17 a full stop is left between "health" and "comments".

-          Please, try not to repeat the same or very similar words too often. E.g. “comments” in line 17, line 18 “it was also commented”, line 20” It was also commented”, line 21 “comments”. Try to find some synonymous.

Keywords: there are two different groups of keywords. Review, please.

Introduction:

-          Lines 37-39: This is the authors’ opinion which is not supported by data or references.

-          Lines 43-46: there are a wide variety of studies supporting this sentence, so, please, support their assertions with previous relevant literature.

Impact of Dengue on Health Systems:

-          Some data on the epidemiology of dengue is needed here, as well as on its impacts on the study population.

Factors That May Aggravate the Clinical Condition of Dengue:

-          It is mentioned that several studies have analysed the factors that worsen the situation, but only one study is cited [9].

-          Lines 101-104: Statement not supported by previous literature.

Physical Activity, Strengthening the Immune System and Dengue: this section is well-structured, well-referenced and comprehensive. it should be the model for redoing previous sections.

-          Line 139, replace (9) by [9].

Final Considerations: A more in-depth analysis as a summary and conclusions of what has been discussed should be made here.

Informed Consent Statement: Should read not applicable as there are no human or animal subjects in this work.

Acknowledgements and Conflicts of Interest must be appropriately worded or deleted.

I think that the proposal needs a little more work to be published, as the subject matter seems interesting and relevant to me. 

Comments on the Quality of English Language

There are no significant comments on the English edition, except for the comments on the Abstract mentioned in the previous section. 

Author Response

Dear Reviewer

Attached our answers

Kind Regards

Francisco Pitanga

Round 2

Reviewer 2 Report

Comments and Suggestions for Authors

Dear authors, 

The text has improved considerably since the first version. However, in response to the request to base the work on a more thorough review of the previous literature, the result has been more self-citation, which is not very recommendable without the requested thorough review. 

Author Response

Dear Reviewer

Thank you more one time for your contribution. Was included in the text two meta analysis and systematic review.

Kind regards